# A Vivid Orange New Genus and Species of Braconid-Mimicking Clearwing Moth (Lepidoptera: Sesiidae) Found Puddling on Plecoptera Exuviae

**DOI:** 10.3390/insects11070425

**Published:** 2020-07-09

**Authors:** Marta Skowron Volponi

**Affiliations:** 1ClearWing Foundation for Biodiversity, 01-866 Warsaw, Poland; m.skowronvolponi@uwb.edu.pl; 2Laboratory of Insect Evolutionary Biology and Ecology, Faculty of Biology, University of Bialystok, 15-245 Białystok, Poland

**Keywords:** insect behaviour, mimicry, mud-puddling, parasitoids, Thailand, barcoding, new species

## Abstract

A clearwing moth with a distinct orange, black and white colour pattern was found sucking up fluids from Plecoptera (stonefly) exuviae on rocks, surrounded by water, on a river bank in Thailand. During this process, known as puddling, the sesiid ejected brown liquid, indicating that it was not imbibing water alone. The behaviour was documented via video recording. Morphological and DNA analyses indicate that the moth is a new genus and species of the tribe Osminiini and it is described herein as *Aurantiosphecia piotrii* genus et species nova. Two species of *Aschistophleps* Hampson, 1893 have been transferred to the newly established genus. The barcode sequence of the cytochrome c oxidase I gene was obtained with universal invertebrate primers after two sets of standard Lepidoptera primers failed to generate a product. Sections on behaviour, conditions of occurrence and possible mimicry models are included. The unique colouration and body posture of *A. piotrii* suggest that it is a braconid wasp mimic, with the mimicry model potentially also being the sesiid’s parasitoid.

## 1. Introduction

Oriental clearwing moths of the tribe Osminiini have been studied from the late XIX century [1], through the early [2] and late [3,4] XX century, until the 2000s [5,6]. These studies have shown that representatives of this tribe are highly diverse in their external and genital morphology, as well as, as indicated in more recent studies, in the model species they mimic, which range from stingless bees [7], through potter [8] and spider wasps [6], to pyrrhocorid bugs [9]. Almost all known species, however, have been described based on pinned specimens only. Nearly nothing is known about the behaviour of the representatives of this tribe, apart from several studies published by the author and colleagues in recent years [7]. This is mostly due to their rarity, elusive nature and common association with unique habitats, as well as difficulties in locating these moths in the field without the use of synthetic pheromones. Many described species are represented by one, often old and destroyed, type specimen alone, with no information on its biology, habitat, behaviour, or even natural body posture. It is surprisingly uncommon for authors describing a new species to identify a group potentially serving as a mimicry model in the evolution of a new taxon of Sesiidae.

The discovery of a peculiar, braconid-mimicking, osminiine species led the author to closely study and compare it to other known taxa. As a result, a new genus and species, *Aurantiosphecia piotrii*
**gen.** et **sp. nov**, is introduced. Two previously described species from Mainland Southeast Asia have subsequently been transferred to the newly established genus.

## 2. Materials and Methods

*A. piotrii* sp. nov. was located and collected without the use of synthetic pheromones. It was filmed and photographed with Sony RX10 II, Sony Corporation, Tokyo, Japan Sony α7R II, Sony Corporation, Tokyo, Japan and Olympus Tough TG3, Olympus Corporation, Tokyo, Japan cameras. Temperature and humidity measurements were taken with an electronic thermo-hygrometer, placed in the shade. The reference specimen was collected at 3 pm on a hot sunny day (33 °C and 51% air humidity), killed with ethyl acetate, and immediately pinned in the field. For DNA extraction and genitalia dissection, the complete abdomen was removed and digested overnight in 55 °C with proteinase K. Further DNA extraction was conducted as described by Knölke et al. [10]. Polymerase chain reactions, with standard Lepidoptera primers [11] for the barcode fragment of the mitochondrially encoded cytochrome c oxidase I gene (MT-CO1), as well as Lep-F1 combined with the reverse primer Enh_LepR1 (5-CTCCWCCAGCAGGATCAAAA-3), failed to generate a product despite performing the PCRs in an annealing temperature gradient. PCRs were then conducted (using the same DNA sample) with LCO1490 and HCO2198 primers [12], which amplified the desired barcode fragment resulting in a 627-nt product. PCR products were Sanger sequenced in an Applied Biosystems 3130 Genetic Analyzer, Applied Biosystems, Waltham, Massachusetts, USA. The sequence has been submitted to GenBank (accession no. MT498852). Genitalia were mounted in Euparal on a standard microscope slide and photographed with a Leica M205A stereomicroscope, Leica Camera AG, Wetzlar, Germany.

## 3. Results

### 3.1. Taxonomic Part

*Aurantiosphecia* Skowron Volponi, **gen. nov.**

Type species: *Aurantiosphecia piotrii* Skowron Volponi sp. nov.

Small clearwing moths (alar expanse 16–22 mm) with a braconid wasp-like appearance and distinct orange colouration on various parts of the body, including wings (Figure 1 and Figure 2). Long, functional proboscis. Conspicuous, upturned labial palpi. Long, skinny, smooth-scaled hind legs with only some elongated scales on the distal part of the hind tibia, resembling hind leg spurs of wasps. Forewings entirely, or mostly, opaque with vein R_5_ missing. Hindwings with opaque distal parts, or almost entirely covered with scales with only the discal area transparent. Representatives of this genus rest with tips of forewings overlapping (Figure 2; Appendix A TC 00:47‒01:16; 02:46‒03:02). Male genitalia with narrow tegumen, long uncus covered with setae, valvae elongated, narrowed in the distal part, densely covered with setae, saccus short. Female genitalia known from *A. murzini* (Gorbunov & Arita, 2002) [13]: with very long ductus bursae, well sclerotized papilla analis and tergite of segment VIII, membranous ostium bursae in segment VIII, membranous antrum with a sclerotized fragment, corpus bursae without signum.


*Differential diagnosis*


This new genus is most similar to *Melanosphecia* Le Cerf, 1916, *Aschistophleps* Hampson, 1893, *Heterosphecia* Le Cerf, 1916 and *Akaisphecia* Gorbunov & Arita, 1995 but can be easily distinguished from all these genera by the absence of conspicuous hind leg tufts of elongated scales (*Aurantiosphecia* gen. nov. has long, wasp-like, smooth-scaled legs with only some moderately elongated scales on the distal part of the hind tibiae), overall body colouration and by the configuration of male genitalia, especially the shape of uncus and valvae. It can be additionally distinguished from *Aschistophleps* by the entirely, or almost entirely, opaque forewings (entirely or with at least some transparency in the anterior, posterior and exterior transparent areas in *Aschistophleps* [3,14,15]), from *Heterosphecia* by opaque forewings, bright orange colouration and the more slender wasp-like body with long hind legs (in comparison to the robust, bee-like representatives of *Heterosphecia* [1,5]), from *Melanosphecia* by the smaller and less robust body and overall colouration (metallic/steel blue colours in *Melanosphecia* [6,16]) and from *Akaisphecia* by the absence of a filiform appendix at the tip of the abdomen [4] and opaque parts of the fore- and hindwings. The male genitalia most closely resemble those of *Chamanthedon* Le Cerf, 1916, especially the distinct long and narrow uncus, but *Aurantiosphecia* gen. nov. differs from this genus significantly: in a functional proboscis; wing venation (veins R_4_ and R_5_ are long and stalked in *Chamanthedon,* vein R_5_ is absent in *Aurantiosphecia* gen. nov.); opaque forewings; orange colouration of various body parts; as well as in the shape of valvae and saccus (which are more rounded in the comparable genus, see Figures 10 and 11 in [17]).

For immediate in-the-field taxonomic placement, apart from the presence or absence of conspicuous hind leg tufts, one could propose establishing a novel genus level diagnostic feature: the position of forewings in a typical resting posture. *Aurantiosphecia* gen. nov. and *Melanosphecia* [6] rest with forewing tips overlapping, whereas the representatives of all other Oriental Osminiini genera keep their wings folded back against their body separately, not overlapping (Figure 2).

*Composition*. The following three species are included in this genus: *A. metachryseis* Hampson, 1895; *A. murzini* Gorbunov & Arita, 2002 and *A. piotrii* Skowron Volponi sp. nov., all of which are braconid wasp mimics.

*Range.* Known from northwestern Myanmar, central Thailandand South Vietnam.

*Etymology*. Aurantio from the Greek word aurantius meaning orange, tawny refers to the vivid mandarin orange scales covering representatives of this genus; sphecia comes from the Greek sphex, meaning wasp, and is derived from the closely related genera *Melanosphecia*, *Heterosphecia* and *Akaisphecia*.


*Aurantiosphecia piotrii*
**sp. nov.**


*Type material***:** Holotype ♂, pinned (Figure 1d). Original labels: “THAILAND: Phetchaburi. 05 III 2017. Skowron Volponi M.A.”; “Holotype, *Aurantiosphecia piotrii* gen. et sp. n., des. Skowron Volponi MA 2020; “Genitalia slide ♂, No. MSV-15”. Will be deposited in the Natural History Museum in London.

Wingspan 18 mm, body length 9.5 mm, antenna 5 mm.

*Head*: antenna covered with black scales dorsally, ventrally brown without scales except for orange-scaled segment margins, minute dense setae on scaleless fragments; vertex with black hair-scales with silver sheen; compound eye black; frons smooth-scaled, black; labial palpus bright orange, upturned, smooth-scaled dorsally, with elongated scales ventrally and apically; proboscis long and functional, bright orange; pericephalic hairs bright orange.

*Thorax*: patagium “collar” bright orange; tegula orange with some black scales; mesonotum bright orange with two dashed black longitudinal stripes and individual black scales here and there; metanotum black, pleuron and sternum black, long bright orange setae at wing insertion.

*Forewing:* entirely opaque with metallic sheen in sunlight, brown and orange with some black: orange scales at base, along costa in basal 2/3, in lower half of cell and along inner margin in basal and discal interspaces, brown with admixture of orange scales in postdiscal and marginal areas, veins brown, patch of black scales from distal part of coastal margin to vein R_3_ and in anterior half of discal cell; discal spot black with some orange scales; cilia dark brown.

*Hindwing:* mostly yellow‒hyaline, transparent areas without scales but space between veins CuA_1_ and CuA_2_, as well as distal parts of spaces between CuA_2_‒1A, 1A‒2A and 2A to anal margin covered with dark brown scales; margins and veins dark brown distally, orange in basal half; discal spot and upper margin of cell orange; cilia quite long, dark brown distally, orange proximally, cover the abdomen in a typical resting position of a live individual (Figure 1 and Figure 2).

*Foreleg:* coxa black mixed with orange and some white scales; femur basal half white, distal half black, smooth-scaled dorsally, with elongated scales ventrally; tibia bright orange with characteristic tufts of hair-like scales ventrally and laterally, dorsally smooth-scaled; tarsomeres bright orange, smooth-scaled with some hair-scales on ventral side of first tarsomere.

*Midleg*: coxa and femur black, smooth-scaled; tibia black dorsally, black mixed with orange ventrally, with slightly elongated scales; spurs orange with some black scales; tarsomeres black dorsally, ventrally alternate bands of bright orange smooth scales and black slightly elongated scales.

*Hindleg:* significantly longer than abdomen; coxa and femur black, smooth-scaled; tibia basally black and smooth-scaled, medially white and smooth-scaled, distally black with elongated scales; spurs black; tarsomeres smooth-scaled, I‒II black exteriorly and white with some orange scales towards the body, tarsomeres III‒V black with some orange scales.

*Abdomen*: dorsally black, tergites III‒VII with narrow white strips on posterior margins; laterally and ventrally white on segments I‒II and proximally III, as well as V‒VIII; distal half of segment III and entire segment IV black laterally and ventrally with narrow white strip on posterior margins; anal tuft very small, black mixed with orange.

*Male genitalia* (Figure 3): tegumen narrowing towards uncus; uncus long and narrow, covered with long setae, narrowing between tegumen and uncus (Figure 3d); gnathos small and narrow; valva elongated, of almost equal width in basal 3/4, narrowed upwards in distal 1/4, densely covered with long hairs in distal 3/4, some specialized setae in basal 1/4; saccus short and narrow, bluntly ended; aedeagus longer than valva by app. 20%, vesica with minute cornuti.

Female unknown.


*Differential diagnosis*


This species is most similar to *A. murzini* and *A. metachryseis*. From *A. murzini* it differs in the presence of an entirely opaque forewing and broader transparent areas of hindwing, colouration of the abdomen (orange dorsally and yellow ventrally in comparable species) and hindlegs (coxa, femur and tibia yellow-orange in comparable species), as well as the length of hindlegs (tarsi II‒V protrude beyond the tip of the abdomen in *A. murzini*, whereas only tarsi IV‒V are beyond the abdomen in *A*. *piotrii* sp. nov.). Compared to *A. metachryseis*, *A. piotrii* sp. nov. differs in the mostly hyaline hindwings (mostly covered with golden yellow and brown scales in *A. metachryseis*), colouration of the forewings (entirely black-brown in *A. metachryseis*, brown and orange in *A. piotrii* sp. nov.), thorax (black in *A. metachryseis*), abdomen (patches of orange laterally in *A. metachryseis*) and hindleg tibiae (mostly orange in *A. metachryseis*), as well as the length of hindleg tibiae (tarsi II‒V protruding beyond the abdomen in *A. metachryseis*).

Specimens examined. Thailand: 1♂ (will be deposited in NHMUK), Phetchaburi, 05.III.2017, MA. Skowron Volponi

### 3.2. Habitat and Conditions of Occurrence

Pebbly river bank with several huts surrounded by mixed plantations on one bank and a semi-evergreen rainforest on the other bank. Only three, or perhaps even two, individuals were seen on three consecutive days in the same location, the holotype was collected on day 2. *A. piotrii* sp. nov. was active on very hot sunny days (31–35 °C and 30%–56% air humidity) in the afternoon, between 1 and 3 pm.

### 3.3. Behaviour

*A. piotrii* sp. nov. was found puddling in unusual circumstances for the family Sesiidae: the moth was sucking up liquids on a rock submerged in flowing water on the edge of a big river (Figure 1a,b; Appendix A TC 00:19–00:22, 00:47–01:20), whereas normally sesiids puddle on moist, but not entirely wet, sand/soil near water puddles or on rocks further away from the stream/river edge. The new species would land on wet rocks without problems (Appendix A TC 02:03–02:24; 02:40–02:44) and sometimes even keep the tips of its hind legs in water during puddling (Appendix A TC 00:19–00:22). A second individual was observed several days later in a similar situation. Other Sesiidae species observed in the same area never landed on rocks surrounded entirely by water. The second observed individual of *A. piotrii* sp. nov. stayed for a long while on a rock with two Plecoptera exuviae (Figure 1b; Appendix A TC 00:47–01:58), licking the moisture around them with its proboscis (it also flew onto another rock with an additional stonefly exuvia). It was very calm and puddled for nearly a full hour, changing rocks from time to time. During this process, it ejected two drops of a brown liquid through its abdomen (Figure 1b; Appendix A TC 01:03–01:20).

*A. piotrii* sp. nov. has long functional hind legs which it uses for locomotion, clinging onto rocks (Appendix A TC 01:49‒01:58) or, along with the fore and mid legs, for taking off and landing (Appendix A TC 01:59‒02:22; 02:41‒02:43). The white colouration of the hind tibia matches the white patches on the sides of the sesiid’s abdomen (Figure 1c), which in flight gives an impression of a “wasp waist” 02:34–02:44), similarly, although not as spectacularly as in *Melanosphecia paolo* Skowron Volponi, 2019. In flight, the skinny legs of *A. piotrii* sp. nov. give it a wasp-like appearance, in contrast to *Aschistophleps argentifasciata* Skowron Volponi, 2018 [15] or *Heterosphecia pahangensis* Skowron, 2015 [18], in which their strongly tufted legs hanging downwards in flight resemble bees.

### 3.4. DNA Barcode Analysis

Interestingly, standard Lepidoptera primers LepF and LepR, and even LepF, in combination with an enhancing reverse primer did not generate a product for *A. piotrii* sp. nov. Folmer primers, known to work well for various invertebrate phyla [12], generated high-quality products. Providing barcode sequences for newly described Sesiidae taxa is unfortunately still a rare event, nevertheless, the slowly growing number of available Osminiini MT-CO1 sequences is beginning to show high intrageneric divergence, reaching 7%–9% [8,18], in this single mitochondrial gene.

Obtaining the MT-CO1 sequence for this new species confirms that it is well separated from representatives of closely related genera. Out of published sequences, the closest match is *M. paolo* with a 14.63% sequence divergence. The only available *Aschistophleps* sequence, *A. longipoda,* shows 17.09% divergence. The author has also recently sequenced MT-CO1 of *A. argentifasciata* which shows a 14.46% divergence from *A. piotrii* sp. nov.; however, only a 575-nt sequence was available for comparison (unpublished sequence). Other Oriental Osminiini representatives had the following levels of MT-CO1 sequence divergence: *H. pahangensis* 15.02%; *H. bantanakai* Arita & Gorbunov 2000 15.48%; *H. tawonoides* Kallies 2003 15.57%; *Pyrophleps vitripennis* Arita & Gorbunov 2000 15.71% and *P. ellawi* Skowron Volponi, 2017 15.76%,

*Etymology.* This species is named after my father Piotr Skowron.

## 4. Discussion

The most distinctive feature of this newly described species is its colour pattern: striking orange colouration of the wings and anterior parts of the body and black in the exterior parts, with flashy patches of white on the sides of the abdomen and tergite margins (Figure 1). Such colour patterns are widespread in Braconidae wasps [19], especially the Agathidinae, Braconinae, Doryctinae and Helconinae and has been termed “the BROW colour pattern”, derived from black, red-orange and white [20]. Agathidinae are all parasitoids of Lepidoptera caterpillars and attack mostly concealed larvae. Although knowledge about their biology is scarce, they have been reported to parasitize Sesiidae larvae [21]. *A. piotrii* sp. nov. has a strikingly similar colour pattern to some of the Oriental representatives of Agathidinae, such as *Zelodia* van Achterberg, 2010 (see Figures 455–457 in [22]), *Amputostypos* Sharkey, 2009 (see Figure 18 in [23]) or *Cremnoptoides* van Achterberg & Chen, 2004 (see Figures 41‒42 in [24]). *A. piotrii* sp. nov. could be both a mimic of and host to an Agathidinae species. Quicke [19] draws a possible explanation regarding the evolution of the host to its parasitoid, arguing that simply their coexistence at the same time and place due to their linked life histories could be the reason—if at least one of the species was unpalatable to predators. Brightly coloured braconids are considered to be aposematic and although the degree of their unpalatability to predators is yet to be determined empirically, some are known to sting and many produce an odour upon attack [19]. *A. piotrii* sp. nov. could thus gain protection from predators as a Batesian mimic, by being part of a homeochromatic assemblage together with Braconidae wasps. One must keep in mind, however, that the palatability of sesiids has also not been studied, thus classifying them as purely Batesian mimics is still only speculative and the true nature of their mimicry is yet to be determined.

The process of sucking up liquids from moist substrate, termed puddling or mud-puddling, has been observed and described by the author for many sesiids [6,8,15,18]. Most of the observed species seem to have been puddling for salt and would lick surfaces where salt solutions were poured [18] or even drink sweat directly from the author’s skin. Puddling for salt is a common habit of different lepidopterans in tropical regions. *A. piotrii* sp. nov., however, was more interested in Plecoptera exuviae (Figure 1b) than moist sand where other sesiids puddled. The brown colouration of waste it ejected has not been observed before in Sesiidae. It is common for moths to eject even large quantities of liquids while puddling for salt (see suppl. video TC 01:27–01:38 of [18]), which is simply excess water [25], however, the waste is usually transparent. The characteristics of insect waste change according to diet [26], the brown colour may indicate that the sesiid was not puddling for water or salt alone, but perhaps for proteins washed out from the exuviae. Beck et al. [27] showed that different families of butterflies in Borneo readily accept solutions of albumin, glycine and lysine. Further testing could evaluate which nutrients diurnal moths such as Sesiidae gain through puddling.

## 5. Conclusions

A new genus and species of braconid wasp-mimicking clearwing moth was found in Thailand and named *Aurantiosphecia piotrii* gen. et sp. nov.The clearwing moth displayed an interesting behaviour of “puddling” i.e., sucking up fluids from Plecoptera exuviae on a river bank and landing on rocks entirely surrounded by water.During the puddling process, the sesiid ejected brown liquid waste, indicating it was not puddling for water and salt alone.Black, orange and white braconid wasps are both potential mimicry models and possibly parasitoids of the new clearwing moth species.Standard Lepidoptera primers failed to generate a product in attempts to amplify the MT-CO1 gene of *A. piotrii* sp. nov. The MT-CO1 was finally obtained with Folmer’s universal invertebrate primers.

## Figures and Tables

**Figure 1 insects-11-00425-f001:**
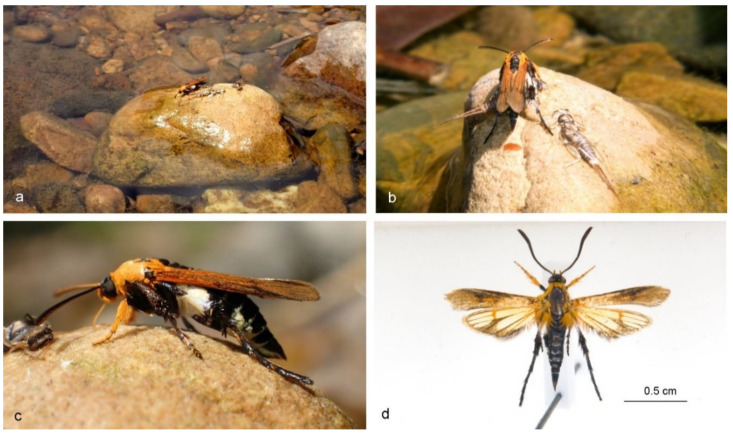
*Aurantiosphecia piotrii***sp. nov**. is an orange, black and white braconid wasp mimic. (**a**) The sesiid was found sucking up liquids on a rock surrounded by water, on a river bank in Thailand; (**b**) It puddled on Plecoptera exuviae and ejected a brown liquid waste during the process; (**c**) *A. piotrii* sp. nov. has long, skinny, wasp-like hind legs and matching white colouration of the hind tibia and abdomen which gives an impression of a “wasp waist”; (**d**) *A. piotrii* sp. nov. holotype, pinned.

**Figure 2 insects-11-00425-f002:**
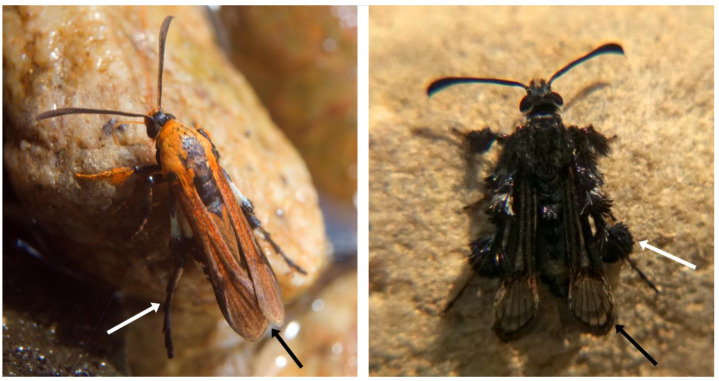
Representatives of *Aurantiosphecia* gen. nov. and *Aschistophleps* Hampson, 1893 can be immediately distinguished in the field. Left: *Aurantiosphecia piotrii* gen. et sp. nov. has long wasp-like hind legs without conspicuous tufts of elongated scales and rests with forewings overlapping. Right: *Aschistophleps argentifasciata* Skowron Volponi, 2018 has strongly tufted hind legs and rests with forewings not overlapping. White arrows indicate differences in the hind legs, black arrows point to differences in the conformation of forewings in a typical resting position.

**Figure 3 insects-11-00425-f003:**
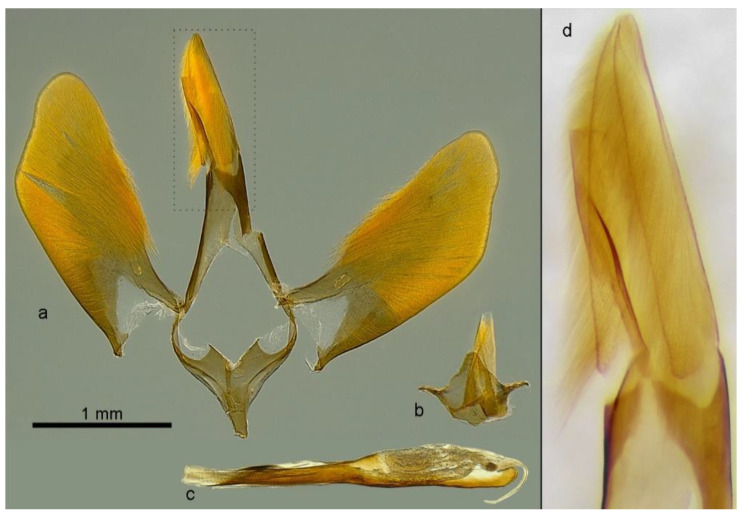
Male genitalia of *Aurantiosphecia piotrii*
**sp. nov.** holotype. (**a**) Ventral view; (**b**) Juxta; (**c**) Aedeagus; (**d**) Details of uncus. Dotted frame marks enlarged area.

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
