# Peer review of "A Vivid Orange New Genus and Species of Braconid-Mimicking Clearwing Moth (Lepidoptera: Sesiidae) Found Puddling on Plecoptera Exuviae"

_insects, 2020, doi:10.3390/insects11070425_

Round 1
Reviewer 1 Report
Dear Author,
This is a very nice and interesting paper that deals with description of a new genus and species as well as some behavior characteristics of the species which is not something that is often observed. I think this paper deserves to be published. Before that I would like to know a few thinks; firstly did you have a permit to collect the species and secondly in the text you discuss that species is puddling on Plecoptera exuviae, but after seeing the video I am not really convinced in that, as they never suck directly from the exuviae itself, it could be that this is not connected what so ever.
Best regards,
Marija Ivković

Author Response
Dear Reviewer,
Thank you for your comments. This study was carried out in Thailand outside of protected areas on unprotected species and thus a permission was not required.
Aurantiosphecia piotrii sp. nov. was observed puddling for a long time on a rock with not one but two Plecoptera exuviae and then flew onto another rock which also had a Plecoptera exuvia on it. In the same habitat there were many other moist rocks to choose from but the moth selected mostly those with exuviae, and there were only two rocks with exuviae in the area. The exuviae were certainly moistened from time to time by the flowing water from the river, thus nutrients were being washed out of them onto the rocks which the sesiid was licking. The brown colouration of the liquid waste that the moth ejected is proof that it was in fact puddling for something different than just water or salt, because in such a case the waste would be transparent, as it is in other puddling sesiids. I do not think A. piotrii landed next to the exuviae by chance and I base these conclusions not only on the observations of this particular species but based on my experience in studying clearwing moths in the tropics, particularly their puddling behaviour. I have added a detailed reference in the manuscript to a supplementary video showing a different species of clearwing moth ejecting transparent liquid waste during puddling (https://vimeo.com/136088402 TC 01:27-01:38).
To answer your question “So was it two or three?”. I deliberately did not give an ultimate answer, because I saw the species three times, three days in a row, but each time a single individual, so I might have seen the same individual two days in a row, whereas the other was collected.
Kind regards,
Marta Skowron Volponi
Reviewer 2 Report
This manuscript provides an exceptionally thorough presentation of a peculiar species of clearwing moths (Sesiidae). This contribution is a rare example of a thorough analysis of specific traits and behaviour, in addition to a standard description of a new genus and species. The attached videos are outstanding, paralleled only with the same author’s earlier ones as far as I have seen. To add to the quality of the videos themselves, the filming conditions must have been close to borders of human endurance.
The manuscript is in general well and fluently written, but as it now stands, the closing sentences of introduction would be more fit as merged into discussion (if needed at all, if my following suggestion seems acceptable). This part of introduction could perhaps be formulated as something such as: “The discovery of a peculiar, braconid-mimicking osminiine species led the author to closely study and compare it to other known taxa. As the result, a new genus and species is introduced”.
I have no objections regarding the systematic conclusions, though with the reservations that speculation of phylogenetic significance of barcode gaps is per se misleading as expressed now in the manuscript, and that mimicry patterns may also be surprisingly misleading regarding the ‘true’ phylogeny. Nevertheless, given the state of current knowledge the solution to erect a new genus for the three species is logical. The discussion section is important in a complicate biological issue such as presented here. It seems to reliably address all the aspects that came to my mind when reading abstract and the earlier parts of the text. I find thus this manuscript of very good quality, but wish that the notes below are considered.
Line 44: I somewhat doubt the relevance of Reference [12] here; it is outdated (and its conclusions later proved partly wrong, even by the authors themselves), and as a lot of new data have been mounting since its publication 2004. I urge the author to search for any more recent references, if any general ones exist; there are numerous case-specific contributions, though. My impression is that the general magnitude of barcode gaps among congeneric species would be smaller than suggested in the reference cited based on the literature I have read. Besides, the more comprehensive taxon sampling within genus, the smaller barcode gaps between species might be expected (and, genera are only subjective units not really reflecting but opinions). I realise that this is not a central issue in the manuscript, but would be good to consider.
Line 81: add brackets to the authors of the species (as it is placed at a different genus than its original one).
Line 109-110: “In genus compared”. Is something missing, as in this form the sentence seems not to make sense?
In diagnoses and descriptions the use of singular/plural in body parts is inconsistent (“papilla analis; forewingS etc.). The general convention is that in diagnoses plurals are allright, but in descriptions solely singular should be used unless specifically needed (such as: valvae basally connceted”).
Line 220 onwards: Barcodes cannot be used as reconstructing phylogenies; they are a phenetic tool suitabale for clustering individual samples. Besides, the larger barcode gap is, the more unlikely it is that barcode would yield a credible deeper phylogenetic signal; the same is though true for other genes as well due to the long-branch attraction phenomenon, or simply because there are no data to condifently link highly divergent taxa anywhere. With the magnitude indicated for Osminiini, there is a serious risk if not even likelihood of getting entirely spurious results with deeper nodes. I suggest the removal or heavy modification of the sentence starting “Unfortunately…” in lines 222-224. That said, I believe that the huge gap to other genera compared are (in)credible enough to support the establishing of a new genus.
I suggest moving the personal part now placed in etymology of the new species to acknowledgements.
Line 240: remove “subfamilies”.
Author Response
Dear Reviewer,
Thank you very much for your thorough and very helpful review. I agree that I may have taken the barcoding discussion a bit too far and have shortened and modified it. I have entirely removed any mention of barcoding from the Introduction section. Section “3.4. DNA barcode analysis” has been modified. Following your suggestion, I tried to find an adequate reference which would give any information on variation in the MT-CO1 sequence in Sesiidae. However, the several studies that have been done on the molecular phylogeny of Sesiidae concern mostly intraspecific variation in North American species and are of little relevance to my discussion (the most recent ones being: Lait LA, Hebert PDN. A survey of molecular diversity and population genetic structure in North American clearwing moths (Lepidoptera: Sesiidae) using cytochrome c oxidase I. PLoS One. 2018;13(8):e0202281. doi:10.1371/journal.pone.0202281;
JA Hansen, WE Klingeman, JK Moulton, JB Oliver, MT Windham, A Zhang, RN Trigiano. Molecular Identification of Synanthedonini Members (Lepidoptera: Sesiidae) Using Cytochrome Oxidase I. Ann. Entomol. Soc. Am. 2012;105(4): 520-528. DOI: http://dx.doi.org/10.1603/AN11028)
I removed the statement “the average MT-CO1 sequence divergence in congeneric species of moths is 6.5% [12]”. You are right that much has changed since the cited paper was published and it is now clear that there is no average value of MT-CO1 divergence that can be credibly given for all diurnal moths, let alone Lepidoptera as a whole. I think this varies greatly even within Sesiidae, with intrageneric divergence not higher than 5% in Synanthedon species (at least those well studied from North America and Europe), but as high as 10% in the Osminiini genera.
The morphological descriptions have been reviewed and singular/plural forms corrected.
I appreciate that you noticed the conditions in which the video was filmed, they were in fact challenging, although the prize highly rewarding.
Kind regards,
Marta Skowron Volponi